# Patient Adherence to Written Instructions following Complete Allergological Evaluation for Suspected Beta-Lactam Allergy: A Tertiary Hospital Study in Greece

**DOI:** 10.3390/jpm13121719

**Published:** 2023-12-17

**Authors:** Michael Makris, Niki Papapostolou, Maria Pasali, Xenofon Aggelidis, Caterina Chliva, Alexander C. Katoulis

**Affiliations:** 1Allergy Unit “D. Kalogeromitros”, 2nd Department of Dermatology and Venereology, Medical School, University General Hospital “Attikon”, National and Kapodistrian University of Athens, 12462 Athens, Greecexaggelides@attikonhospital.gr (X.A.);; 22nd Department of Dermatology and Venereology, Medical School, University General Hospital “Attikon”, National and Kapodistrian University of Athens, 12462 Athens, Greece; akatoulis@med.uoa.gr

**Keywords:** beta-lactam antibiotics, allergic reactions, adherence to instructions, drug provocation tests, patients’ education, compliance, delabelling

## Abstract

Background: Beta-lactam (BL) antibiotics are among the most prescribed groups of drugs worldwide and have been implicated in a variety of allergic reactions. There is a paucity of literature regarding patient adherence to prescribed instructions following comprehensive allergy assessments. Objective: The objective was to follow up the clinical course of BL allergy in patients who underwent thorough allergological investigation for suspected BL allergy at a tertiary hospital and ascertain patients’ compliance with the provided written instructions. Materials: An observational study in patients referred for suspected BL allergy who underwent a comprehensive allergy workup (in vivo ± in vitro tests, DPT in culprit and/or alternative BL) and who subsequently received written instructions was conducted. Data on the nature of the reported drug hypersensitivity reaction, the culprit BL drug, the allergological workup, and the detailed instructions provided in a written drug allergy report were collected retrospectively. Patients’ compliance with the instructions was recorded by a telephone survey using a pre-defined questionnaire. Results: Among the 212 patients meeting the inclusion criteria, 87 patients (72.4% women; mean age 50.1 years; age range 6–84 years) responded to the telephone survey and were included in this study. Surprisingly, 45 out of 87 (51.7%) patients did not adhere to the written instructions. The primary factor contributing to non-compliance was the fear of re-occurrence of a drug-induced allergic reaction (personal and/or triggered by their treating physician reluctance), accounting for 77.7% of cases. The analysis demonstrated that the initial reaction’s severity and type, as well as the outcomes of skin testing, did not correlate with compliance to instructions (*p* > 0.05). Surprisingly enough, a drug provocation test (DPT), irrespectively of the result, emerged as a negative predictor for adherence, with only 40.6% of DPT patients complying compared to 77.8% of those who did not undergo DPT (*p* = 0.005; odds ratio = 0.195; 95% confidence interval: 0.058–0.655). Variables such as performing DPT with alternative or incriminated drugs or the result of the DPT (positive–negative) were not associated with patient compliance. Conversely, the type of instructions provided exhibited a noteworthy correlation with compliance. Patients who were explicitly instructed to entirely avoid all BL antibiotics demonstrated markedly higher adherence rates (83.3%) compared to those who were advised to have a partial or complete release of BLs (31.8% and 58.1%, respectively; *p* < 0.05). Notably, among compliant patients who received either the original culprit drug or the alternative (32 out of 87, 36.7%), no allergic reactions were reported. In contrast, among the 12 patients with written avoidance of all BLs, subsequent BL intake led to immediate reactions (Grade I and IV) in 2 patients (16.6%). Conclusions: A notable disparity in patient adherence to written instructions prohibiting or releasing beta-lactams was demonstrated. Less than half of the patients ultimately complied with the provided instructions, underscoring the need for tailored patients’ education and strategies to improve adherence in the management of suspected BL allergy.

## 1. Introduction

Beta-lactam antibiotics (BL) represent a widely prescribed class of drugs globally and, along with nonsteroidal anti-inflammatory drugs (NSAIDs), are frequently implicated in various kinds of allergic reactions [1]. Approximately 10% of the general population declare some form of BL allergy, with even higher incidence being reported among hospitalized patients [2]. Paradoxically, subsequent to allergy investigations, up to 90% of patients exhibit negative test results and demonstrate tolerance to BLs [3,4,5]. On the other hand, the designation of BL allergy carries negative socioeconomic implications due to the potential utilization of less efficacious, more toxic, more expensive, and bacteria-resistant alternative antibiotic categories [6,7]. The implementation of delabelling initiatives has gained global attention, aligning with the objectives of antimicrobial stewardship programs (ASPs) [8].

Furthermore, it is noteworthy that individuals allergic to penicillin may paradoxically exhibit tolerance to alternative BLs and vice versa [9]. A fundamental shared attribute between penicillin and beta-lactam antibiotics is the presence of a common beta-lactam ring. However, distinctions arise in the configuration of the adjusting ring and the composition of side chains (R1 and R2) [5]. Less than 2% of patients with positive skin tests to penicillin reagents manifest reactions to cephalosporins [10]. In cases of selective aminopenicillin allergy, cross-reactivity rates escalate to 25–35% due to the existence of shared or identical R1 side chains between aminopenicillins and certain cephalosporins. For those exclusively allergic to aminopenicillins, allergological evaluation offers the potential for treatment options with other BLs, specifically cephalosporins with distinctive side chains [11].

While a crucial need to comprehend the complex factors accompanying penicillin allergy acquisition exists, an equally vital process entails recording the consequences of patient compliance subsequent to the delabelling process. Regrettably, data pertaining to patient adherence to instructions following thorough allergy evaluations are scarce. In a retrospective analysis of patients with multiple drug allergy labels, including beta-lactams (BLs), approximately one third of patients underwent a course of penicillin over the year following the delabelling process. Notably, almost 65% expressed a willingness to consume the delabelled drugs in the subsequent follow-up survey. While not statistically significant, psychiatric comorbidities were observed in 43% of this cohort, aligning with prior research indicating that anxiety and depression may contribute to an increased number of allergy labels [12,13]. These factors are likely influential in determining compliance rates. Furthermore, the presence of a genuine penicillin allergy label has demonstrated a significant association with greater adherence to antibiotic guidelines in two English National Health Service (NHS) trusts, as compared to non-allergy labels [13].

The aim of the present study was to evaluate the long-term outcomes of a thorough BL allergological evaluation, focusing on compliance rates to written instructions and subsequent clinical outcomes of the re-administration of BL antibiotics. Moreover, we assessed patients’ perception of reasons for compliance or non-compliance with drug allergy reports, along with possible risk factors associated with compliance rates.

## 2. Methods

### 2.1. Study Design

We conducted a single center observational study of patients referred to the Allergy Unit “D. Kalogeromitros” of the 2nd Dpt of Dermatology and Venereology in Attikon University Hospital in Athens, Greece, with suspected allergy to BL who completed a full allergological workup (in vivo and/or in vitro tests and/or DPT in the culprit and/or an alternative BL antibiotic) and received a written drug allergy medical report with detailed instructions for the future use of BLs [9]. This study was conducted from January 2021 to February 2023.

We retrospectively recorded data from the medical records—in harmonization with the ENDA/GAHD and GA^2^LEN questionnaire—referring to the following parameters: the nature of the allergic reaction, the implicated culprit drug, the allergological workup, and the personalized detailed instructions documented in the issued drug allergy report [14]. In order to estimate patients’ compliance with the instructions, we designed a structured questionnaire which was addressed to all study participants via telephone survey and was filled by the doctors of the Allergy Unit (Appendix A). The inclusion criteria were (i) suspected BL allergy (immediate or non-immediate), (ii) completion of the recommended full allergological workup, (iii) provision of a written drug allergy medical report, and (iv) informed consent to take part in the telephone survey. Patients who did not complete the investigation or did not receive written instructions for any reason were excluded from this study. In addition, patients who were advised to use BLs in the written drug allergy medical report, but did not require intake at the time of the survey, were also excluded from this study.

“Culprit” BL antibiotic was defined as the one implicated in the initial allergic reaction based on patients’ documented medical history. Conversely, an “alternative” BL antibiotic denoted any other BL that yielded negative evaluation results, subsequently confirmed by drug provocation test (DPT).

The classification of immediate hypersensitivity reactions (IHRs) and non-immediate hypersensitivity reactions (NIHRs) to BLs followed the timing of the index reaction, adhering to the European Network for Drug Allergy classification of drug hypersensitivity reactions [15]. Furthermore, the severity of IHRs was graded according to the World Allergy Organization (WAO) systemic allergic reaction grading system [16].

Patients who were advised to reuse all or some BLs and avoid others were considered compliant when they had received at least one of the suggested BL antibiotics when they required intake. Among patients with confirmed BL allergy who were instructed to avoid all BLs, compliance was determined by the absence of any BL antibiotic usage.

### 2.2. Allergological Work-Up

Briefly, the allergological evaluation included skin prick tests (SPTs), followed by intradermal tests (IDTs) utilizing penicillin commercial reagents, specifically benzylpenicilloyl-poly-L-lysine (PPL); minor determinants mixture (MDM) (Diater^®^, Madrid, Spain); and commercial formulations of penicillin G, amoxicillin, ampicillin, cefuroxime, cefaclor, cefprozil, and any other implicated BL antibiotic based on each patient history.

The concentrations employed for skin testing adhered to those recommended by the European Network for Drug Allergy (ENDA) position paper [9]. Exclusively for cephalosporins, SPTs/IDTs were performed with intravenous formulations of the drugs under investigation. If not available, SPTs were carried out using a pulverized tablet or syrup.

In vitro investigation with the measurement of specific IgE (sIgE) levels (Inmunocap^®^, Phadia, Uppsala, Sweden) to penicilloyl V, penicilloyl G, amoxicilloyl, ampicilloyl, and cefaclor was carried out in all subjects with immediate reactions. In cases of uncertain or delayed reactions, SPTs/IDTs and patch tests with commercial tablets diluted to 10% and 30% in paraffin were conducted, with both immediate and delayed readings at 48 and 72 h. In the context of skin testing, a positive result was defined by a wheal measurement equal to or exceeding 3 mm after 15 to 20 min, with reference to a positive histamine control and a negative normal saline SPT result. Conversely, a positive IDT result was defined as a wheal increase of 3 mm or more from baseline, observed 20 to 30 min following intradermal injection of 0.03 to 0.05 mL of the drug; a negative ID saline control was also carried out [17].

After the in vivo and/or in vitro evaluation and risk stratification, a single-blind, placebo-controlled DPT was performed; the cumulative therapeutic dose was administered in the Allergy Unit under observation in case of immediate reactions and the treatment was continued for 3 days in cases of delayed reactions.

For patients with no definite history of anaphylaxis who tested negative in both in vivo and in vitro tests, the DPT was administered using the culprit BL antibiotic. By contrast, patients exhibiting selective aminopenicillin or cephalosporin allergy underwent a DPT with an alternative BL antibiotic, based on positive SPT/IDT and/or sIgE measurements. Patients allergic to the beta-lactam ring, as indicated by history and positive SPT/IDT and/or sIgE measurements, were advised to strictly avoid all BL antibiotics and were informed of the option of rush desensitization in a specialized allergy center in a hospital setting if the offending drug was considered irreplaceable. All patients gave written inform consent for DPT.

All collected data were obtained for clinical purposes, and no identifiable information was available to clinicians who were not part of the clinical team. Ethical approval was obtained from the Ethics Committee of University General Hospital “Attikon” (number 312/2-1-2021) despite this study’s observational, non-interventional nature. All medical procedures were conducted according to Good Clinical Practice guidelines and informed consent was obtained from all patients to participate in this study.

### 2.3. Statistical Analysis

Data are presented as numbers and percentages for nominal parameters and as mean and SD for continuous variables. Comparisons between 2 groups were performed with Student’s *t* test or Mann–Whitney U for continuous parametric or non-parametric variables and with a chi-squared test or Fisher’s exact test for categorical variables, each when appropriate. Logistic regression analysis was used to assess the independent effects of various risks factors on patients’ compliance. All tests of hypotheses were considered significant when 2-sided probability values were *p* < 0.05. Statistical analyses were performed with IBM, SPSS-24 (Armonk, NY, USA), and Graph Pad Prism 9 was used for the design of the graphs.

## 3. Results

### 3.1. Patient Demographics

A total of 356 patients underwent evaluation at the Allergy Unit for suspected BL (beta-lactam) allergy. Of these, 212 patients met the first three predefined inclusion criteria. Telephone calls were made to all 212 eligible subjects and, finally, 134 out of them agreed to participate and provide responses to the questionnaire. Based on the structured questionnaire, 47 patients did not require BL intake and were consequently excluded from this study. In total, 87 patients were included in this study (67 females (72.3%), mean age 50.13 years (±19.5 years, range 6–84 years)). The demographic characteristics of the participants are comprehensively presented in Table 1.

### 3.2. Index Reaction and Allergological Work-Up

The index reaction was classified as immediate hypersensitivity reaction (IHR) in 59 patients (67.8%), while the remaining 28 patients (32.2%) reported non-immediate hypersensitivity reactions (NIHR). IHRs according to the World Allergy Organization (WAO) grading system were classified as 50.8% (Grade I), 11.9% Grade II, 11.9% Grade III, 11.9% Grade IV, and 13.6% Grade V. Referring to NIHRs, a maculopapular exanthema was identified as the most prevalent type, affecting 92.8% of the patients.

Skin tests (STs) were performed on 78.2% of the patients, yielding positive results in 31% of cases. The average time interval between the reaction occurrence and the execution of allergy workup was 43.4 months (±89 months). For patients deemed to be at “low risk” based on their clinical history, a direct drug provocation test (DPT) was conducted without preceding ST. Conversely, STs were omitted for patients who experienced severe index reactions (Grade III–V) and exhibited positive in vitro findings indicative of BL allergy (BL ring allergy).

### 3.3. Drug Provocation Tests

A DPT was performed in 69 patients. Among them, 43.5% underwent DPT with the culprit drug to eliminate the BL allergy label. Notably, 26 out of 30 patients successfully passed the DPT, leading to the removal of the BL allergy label, while 4 patients experienced mild reactions (Grade I or II). In the remaining 56.5% of cases, an alternative BL antibiotic was subjected to DPT based on ST and/or in vitro results, as well as individual patient risk assessment. In total, an uneventful DTP was observed in 92.6% of cases, whereas 7.4% experienced only mild reactions.

### 3.4. BL Allergy Label

Following comprehensive allergological evaluation, every patient received a drug allergy report with specific instructions for future use of beta-lactams. The BL allergy label was fully removed in 35.6% of patients, while allergy was confirmed in 13.8% of cases. The remaining 50.6% of patients exhibited allergies to aminopenicillins or specific cephalosporins, but demonstrated tolerance to alternative BL antibiotics. (Figure 1) The tolerance was affirmed through negative DPT outcomes.

### 3.5. Compliance with the Drug Allergy Report

In total, 48.3% adhered to the drug allergy report. (Figure 2) This compliance entailed both the avoidance (23.8%) and release of BL use (33.3% partial release, 42.8% complete delabelling). Notably, no allergic reactions occurred in the whole population of compliant patients based on the phone survey.

Conversely, 51.7% of patients deviated from the prescribed instructions, as confirmed by the telephone survey. Within this group, 28.8% had their BL allergy label removed. Among non-compliant patients, 4.4% were advised to fully avoid BLs, while in 66.6%, the partial release of certain BLs was advised. Fear for possible allergic reactions from the patient and/or the reluctance of treating physicians (77.8%) accounted for the vast majority of the lack of compliance. Other reasons included BLs not being the chosen antibiotic for the patient’s specific condition (e.g., prostatitis, urinary tract infection, skin infection) (20%), while in 2.2%, it remained unspecified. (Figure 3).

Notably, in 2 out of 12 patients who received a written report for the strict avoidance of all BLs, after the subsequent intake of BL by mistake, they both experienced an immediate reaction (Grade I and Grade IV, respectively) The first case involved a 42-year-old female who initially exhibited a Grade IV reaction to amoxicillin. The in vivo workup yielded positive skin tests (STs) to penicilloyl-polylysine (PPL), minor determinant mix (MD), amoxicillin, cefuroxime, and cefaclor, leading to the prohibition of all BLs. Approximately one year after the evaluation and receipt of a written drug allergy medical report, the patient, without a doctor’s prescription, took cefuroxime tablets for a urinary tract infection and experienced a severe Grade IV reaction characterized by urticaria, dizziness, and loss of consciousness. In the second case, a 50-year-old male experienced an initial Grade IV reaction to amoxicillin/clavulanic acid. The in vitro evaluation showed positive results (c1: 2.61 kU/L, c2: 2.65 kU/L, c5: 0.35 kU/L, c6: 2.87 kU/L, c7:0.27 kU/L), with a total IgE of 250 IU/mL, leading to the recommendation to avoid all beta-lactams. After 23 months, the patient received cefaclor tablets from his dentist for dental inflammation, resulting in a mild Grade I immediate reaction characterized by urticaria and angioedema.

No significant variations in patient compliance were identified when comparing reaction type and the severity and performance of STs. (Figure 4a–b) Furthermore, no disparities emerged when analyzing DPT performance, DPT outcomes, or the specific type of BL (alternative or suspected culprit) used in the DPT. (Figure 4c–e) However, a distinct correlation was established between instruction type and patient compliance. Specifically, patients instructed to avoid all BLs exhibited notably higher compliance rates (83.3%) compared to those that were advised to use standard alternative BLs (58.1%) or were fully relabeled (31.8%) (*p* < 0.05) (Figure 4f).

Surprisingly enough, undergoing a DPT emerged as a negative predictor for adherence, with only 40.6% of DPT patients complying compared to 77.8% of those who did not undergo DPT (*p* = 0.005; odds ratio = 0.195; 95% confidence interval: 0.058–0.655). Regarding the outcomes of the DPT, a positive DPT was linked to a 20% compliance rate and an 80% non-compliance rate, whereas a negative DPT was associated with a 41.3% compliance rate and a 58.7% non-compliance rate. Although there was a trend suggesting higher non-adherence with a positive outcome, it did not reach statistical significance.

Using the same variables as above in a multivariate logistic regression analysis, we identified no risk factors regarding compliance to written instructions.

## 4. Discussion

In the realm of the antibiotic allergy epidemic, with penicillin and BL allergy constituting a public health issue, the process of beta-lactam (BL) allergy delabelling holds significant importance in the fields of medicine and patient care [18,19]. The delabelling of BL allergy is a vital process that ensures accurate allergy status, expands treatment options, aids in antibiotic stewardship, reduces antibiotic resistance, alleviates patient anxiety, and can lead to significant cost savings. However, even after the label has been removed, patients’ compliance post-evaluation remains to be determined, as the label may persist after the evaluation or patients may refuse to use BLs due to fear and ignorance. In addition, physicians may tend to be reluctant to give BLs even after a negative evaluation, highlighting the problem of the BL allergy label even more [20,21].

In the present study, the BL allergy label was completely removed in more than 1/3 of patients (35.6%) and 50% could tolerate at least some BL antibiotics. Allergy to penicillin and other BLs antibiotics was confirmed in 13.7%. These findings are consistent with those of other studies and demonstrate the need for comprehensive allergy evaluation [22,23]. However, the post-evaluation assessment of patients’ compliance with written instructions reveals a significant concern. The observed non-compliance rate among patients, surpassing half of the study cohort, emphasizes the challenges associated with instructing patients in managing BL allergy.

The primary driver of non-compliance in this study, as indicated by patients’ responses to a questionnaire, was fear of experiencing new allergic reactions. This underscores the emotional and psychological factors that influence patients’ adherence to medical advice. Furthermore, it is crucial, and has already been suggested, that instructions regarding drug allergy should be clear and not overly detailed. Hence, providing an allergy passport or a comprehensive certificate report has been suggested for every patient with drug allergies [24].

In the realm of that, the observed low compliance rates in the present study appear to be closely tied to the type of instructions provided to patients, introducing a critical dimension for comprehending patient behavior. Patients who were advised to completely abstain from all beta-lactam (BL) antibiotics and were labeled as allergic to all BLs demonstrated significantly higher levels of adherence compared to those instructed to resume their use partially or fully. In the last ones, the BL allergy label was partially or fully removed, respectively. This observation underscores and supports literature data regarding the profound influence of instruction clarity and the patient’s perception of risk on adherence [24].

Based on the above-mentioned observation, it seems easier for patients to accept the label of BLs rather than the delabelling procedure. The suggestion of resuming or using an alternative BL antibiotic, even after a negative DPT, tends to encounter resistance from patients, primarily due to apprehension. This fear can stem from either personal concerns or guidance from their treating physician, emphasizing the necessity for more informative training programs regarding drug allergies for primary care physicians.

Although it is not clear from our study whether the fear reported by patients was personal or associated with physicians’ perspectives, the literature reveals a reluctance trend among doctors to prescribe antibiotics for patients who report an allergy despite a medical report of a negative DPT [25].

Results from a large prospective study focusing on penicillin allergy within surgical populations support this observation: less than 50% of attending physicians were willing to administer penicillin to patients who had been delabelled by an allergy specialist employing a DPT. Conversely, an intriguing discovery was that certain physicians demonstrated a willingness to administer penicillin even when a specialist had designated a penicillin allergy label if they believed the label was inaccurate [25]. This observation raises concerns about potential patient safety issues associated with label inaccuracies. Notably, our own study further underscores this concern, as it witnessed an instance where 2 out of 12 patients, advised to avoid beta-lactams (BLs), did not adhere to the instructions and, consequently, experienced immediate allergic reactions. Although the first patient received the BL antibiotic on his own without a prescription, the second one had a prescription from his dentist, despite the drug allergy medical report.

While DPTs remain the gold standard for diagnosing and managing drug allergies, it is intriguing and rather surprising to note that our study observed a lower adherence rate among patients undergoing DPTs. Although our research design did not aim to elucidate the causative factors behind this observation, it could be associated with the stress experienced during the DPT procedure, or it may be a random finding that necessitates validation through larger-scale studies. We believe that the performance of a DPT in a hospital setting provides safety and certainty to patients, and while they are willing to intake the drug during the procedure in a hospital setting, they are reluctant to do so in a real-life setting where they might feel more vulnerable to a reoccurrence of allergic reactions.

To tackle the adherence issue, a collaborative approach across multiple healthcare centers is crucial. Developing educational materials, including interactive tools like quizzes, reminders, and FAQs, or incorporating apps for patients with drug allergies is essential. These materials should consider the severity of the initial reaction and individual risk profiles, providing clear and personalized information to enhance patient understanding and alleviate concerns. Counseling programs, ideally featuring one-on-one discussions between patients and healthcare professionals, can effectively address misconceptions and fears. Emphasizing the importance of adherence by clearly communicating the risks associated with non-compliance is vital. Establishing a feedback mechanism for patients to report difficulties or concerns, empowering them to actively participate in healthcare decisions, and scheduling regular follow-ups are additional measures to enhance adherence. By implementing a combination of these strategies, tailored to the specific needs and concerns of individual patients, there is potential to significantly improve adherence to written instructions in managing suspected beta-lactam allergy.

As with any study, certain limitations must be acknowledged. The retrospective nature of the study design and the reliance on telephone surveys for compliance assessment introduce the potential for recall bias. Additionally, this study’s focus on a single institution may limit the generalizability of its findings. Future research could incorporate larger, multicenter studies and explore long-term outcomes following DPT-based label removal.

Despite its limitations, this study contributes significantly to the body of knowledge surrounding BL allergy management and patient compliance. The results underscore the need for tailored patient education interventions that address fears and anxieties related to subsequent allergic reactions. Furthermore, the negative association between DPTs and adherence suggests the importance of thorough patient counseling and psychological support before and after such testing.

Future research endeavors could delve into exploring strategies to enhance patient understanding, trust, and cooperation, with a specific focus on addressing the apprehension surrounding DPTs and minimizing the perceived risks associated with BL allergy management. Additionally, larger multicenter studies could provide a broader perspective on patient behaviors and attitudes towards compliance in diverse clinical settings.

## 5. Conclusions

In conclusion, this study highlights a significant disparity in patient compliance with written instructions regarding beta-lactam (BL) antibiotic use post-allergy assessments. Surprisingly, more than half of the patients did not adhere to the provided guidance, primarily due to the fear of allergic reactions. In the current landscape of a drug allergy epidemic, it is crucial to not only emphasize the delabelling process to address this issue, but to also focus on tailored patient and physician education post-evaluation. Ensuring adherence to instructions provided in a drug allergy medical report after a comprehensive allergological investigation is vital, maximizing the benefits of delabelling or appropriate labelling processes.

## Figures and Tables

**Figure 1 jpm-13-01719-f001:**
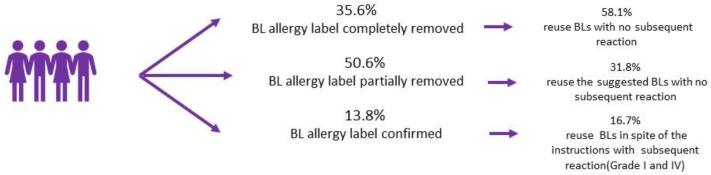
Reuse of BLs based on the delabelling process.

**Figure 2 jpm-13-01719-f002:**
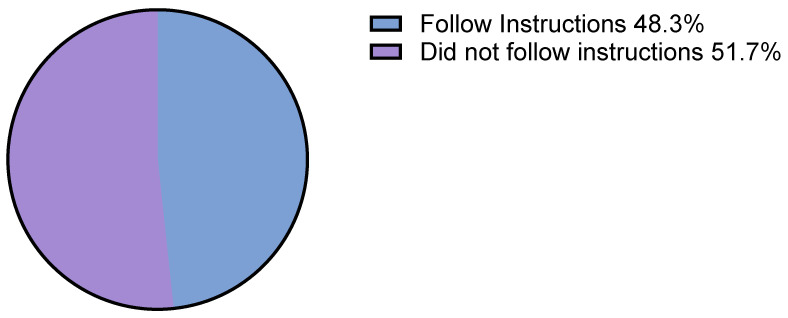
Patients’ compliance to written drug allergy medical report.

**Figure 3 jpm-13-01719-f003:**
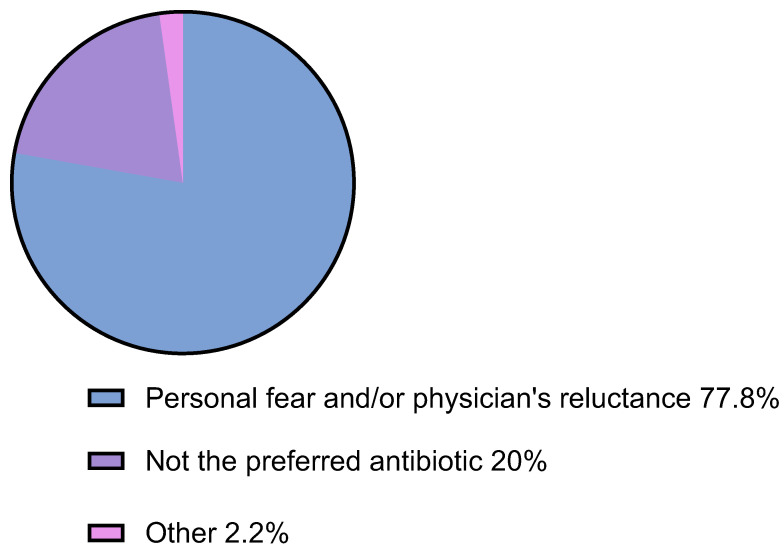
Reasons for not receiving BLs.

**Figure 4 jpm-13-01719-f004:**
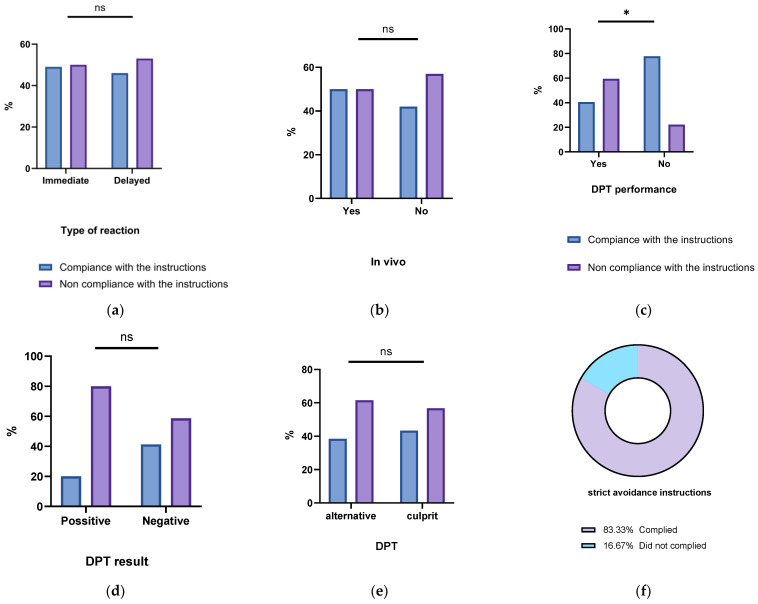
(**a**–**f**) Compliance and non-compliance rates based on: (**a**) the type of the reaction, (**b**) in vivo investigation performance, (**c**) DPT performance, (**d**) DPT result, (**e**) DPT to alternative or culprit BL and (**f**) Compliance rates in strict BLs avoidance instructions. NS: nonsignificant and * means *p* < 0.05.

**Table 1 jpm-13-01719-t001:** Characteristics of the patients.

Variables	Total
Total	87
Sex	
- Female	63 (72.4%)
- Male	24 (27.6%)
Age (years)	50.1 ± 19.5
- Mean ± SD	Range (6–84 years)
Type of reaction	
Immediate	59/87 (67.8%)
Delayed	28/87 (32.2%)
Severity of the index reaction (immediate)	
Grade I	30 (50.8%)
Grade II	7 (11.9%)
Grade III	7 (11.9%)
Grade IV	7 (11.9%)
Grade V	8 (13.6%)
Antibiotic that caused the reaction	
Amoxicillin	20 (23%)
Amoxicillin/clavulanic acid	28 (32.2%)
Penicillin	1 (1.1%)
Piperacillin/tazobactam	4 (4.6%)
Cefuroxime	11 (12.6%)
Cefprozil	6 (6.9%)
Cefaclor	10 (11.5%)
Other	17 (19.5%)
Time between index reaction and evaluation (months)	43.4 ± 89
In vivo	68/87 (78.2%)
Immediate	47/59 (79.7%)
- Positive	21/47 (44.6%)
Delayed	21/28 (75%)
- Positive	6/28 (21.4%)
DPT	69/87 (79.3%)
Culprit	30/69 (43.4%)
Alternative	39/69 (56.5%)
DPT result	
Positive	5/69 (7.2%)
Negative	64/69 (92.7%)

## Data Availability

Data are available on request due to restrictions.

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
