# Peer review of "Patient Adherence to Written Instructions following Complete Allergological Evaluation for Suspected Beta-Lactam Allergy: A Tertiary Hospital Study in Greece"

_jpm, 2023, doi:10.3390/jpm13121719_

Round 1

Reviewer 1 Report

Comments and Suggestions for Authors

The authors have provided comprehensive view of patient adherence to written instructions following suspected beta-lactam allergy. The manuscript is well detailed, and needs minor formatting changes as under:

1.       On line 228 please incorporate units in the same line using non-breaking spaces and on line 238: please put percentage and p value before the fullstop.

2.       What additional strategies that could be proposed to enhance patient adherence to written instructions.  Please include this information in the discussion section. Thank you.

Author Response

please find the reply in the attached document below.

Reviewer 2 Report

Comments and Suggestions for Authors

It is a well designed study. If it would be made prospectively, that would be more valuable.

Author Response

  • Response: Thank you for your favorable consideration of our manuscript. While conducting the study retrospectively was necessary due to the nature of our focus on compliance rates after allergological evaluation, we acknowledge the potential value of a prospective design. Future initiatives could explore this approach to enhance the study's overall robustness and applicability.

Reviewer 3 Report

Comments and Suggestions for Authors

This work, described in the manuscript, will be of interest to both clinicians and those involved in basic allergy/immunology. To improve the perception of the manuscript, authors need to make adjustments:

- Author affiliations should be added to the manuscript.

- The introduction is too short and does not address the entire issue described in the objective.

- Perhaps Figs. 1-3 can be represented inside bar charts or pie charts?

- Line 92: of the Allergy Unit.(Appendix 1) → of the Allergy Unit (Appendix 1).

- In the Study design chapter, it is necessary to add the rationale for the sampling in this study. Is 87 patients sufficient?

- Where are the links to the ethics committee? Have patient consents been obtained?

Author Response

please see the attached document below.

Round 2

Reviewer 3 Report

Comments and Suggestions for Authors

The authors of the manuscript made adjustments to the text.